# Moral luck in investment contexts: We consciously find unprofitable investments less moral

Raphael Max[1]*, Matthias Uhl[2]

**1** TUM School of Social Science and Technology, Technical University of Munich, Munich, Germany,
**2** Faculty of Computer Science, Technische Hochschule Ingolstadt, Ingolstadt, Germany

* r.max@tum.de

## Abstract

Moral luck refers to whether an actor is morally praised or blamed for an action whose outcome they could not influence. In two studies, we investigated the behavioral importance of this phenomenon in the realm of investments, which has become increasingly subject to ethical evaluations. In our first online experiment, we examined whether people's moral evaluation of an investment decision depended on its arbitrary outcome and whether their interpretation of the nature of the decision was driven by this outcome. Our results showed that profitable investments were considered more moral than unprofitable investments. Moreover, profitable investments were labeled "investments" instead of "speculation" or "gambling" more often than unprofitable ones. In our second study, we asked the subjects to assess investments independent of the outcome. After the outcome was announced, the subjects were given the opportunity to reflect and change their initial decision. The results show that people change the moral evaluation and label of investments when told that it had a bad outcome. This observation was stable across different investment contexts. These findings suggest that we must be careful with the increasing moralization of investment decisions and be sensitive to our cognitive biases.

**Data Availability Statement:** All relevant data are within the manuscript and its Supporting Information files.

**Funding:** The authors received no specific funding for this work.

## Introduction

In the wake of the 2008 financial crisis, actors in financial markets were accused of immoral actions. This moral judgment was driven by consequentialist considerations that emphasized the negative impact of the individual and collective economic decisions made by financial intermediaries, bankers and investors. In this sense, it was largely an ex post judgment. However, this raises an important question: why were investors not criticized for immoral behavior in the years before the economic crisis unfolded? In other words, is an investor who took a risk and was lucky more moral than an investor who took the same risk but was unlucky?

Williams [1, p. 241] defines the term "moral luck" as a puzzle because it refers to cases where the agent is morally evaluated although it is clear that a significant factor that determined the outcome of their actions was beyond their control. Nelkin [2] argues, "Immunity

**Competing interests:** The authors have declared that no competing interests exist.

from luck has been thought by many to be part of the very essence of morality" [para. 1]. However, behavioral research has provided evidence that people base evaluations of misbehavior on the severity of the moral outcome. Bazerman and Tenbrunsel [3] use an example of a medical scientist who manipulates research data to get a drug approved. Some participants were told that the drug has lethal side effects, whereas other participants were told that it is safe. In a between-subjects comparison, the scientist's data manipulation was considered significantly more moral if the drug caused no lethal side effects. When participants were confronted with both stories simultaneously, they did not think that a moral difference should be made, indicating that people did not think that the side effects *should* make a difference.

Somewhat contradictory, many people seem to believe that arbitrary consequences should matter for the blame or punishment that we attribute to an agent. Arguably, few people would punish a drunk driver who was lucky enough to get home without incident as severely as an equally drunk driver who runs over a child. The belief that even highly arbitrary consequences matter substantially when attributing punishments is an essential principle of jurisprudence, not an unconscious outcome bias. This is underlined by the fact that in most of the world's legal systems the partly arbitrary effects of a crime often have a substantial influence on the criminal's punishment [4].

Given the increasing discussion surrounding sustainable and ethical investments, it is important to understand whether a person's moral evaluation of an investment is driven by arbitrary monetary outcomes or is independent of these, as all definitions of ethical investments seem to imply. Even when a moral evaluation of an investment actually hinges on its monetary outcome, it is still an open question whether people commit a fallacy according to their own standards because people's intuitions may differ from experts' views. In this respect, it is crucial to understand whether outcome-based evaluations are the result of unconscious bias or people's moral reasoning in the domain of investments simply being consequentialist (i.e., whether they think that an investment's success or failure should actually determine its moral quality).

For several decades now, many asset classes have described themselves as ethical, social, green or sustainable. Sustainable, responsible and impact investment (SRI) and environmental, social and governance (ESG) are common labels for investments. Due to the multitude of normative concepts used in this regard, we use the term ethical investment as a common denominator for different approaches prevailing in the discourse, such as ESG, SRI or sustainable investment. What do we mean by ethical investments? Investment decisions in general and interest in particular have always been the subject of ethical scrutiny, as evidenced by the church's ban on interest or halal investments in Islam. However, the discussion about ethical investment has recently (re-)entered public discourse and observers' way of defining ethical property varies strongly. According to Sparkes [5], churches in the USA and UK have imposed "ethical" restrictions on investment decisions since the early 20th century. Unfortunately, no uniform definition of "ethical investments" exists. One definition pertains to integrating social, environmental or governance criteria in professional investor's calculus [6–8]. Others define "ethical" as refusing to invest in companies that make their money with alcohol, pornography, tobacco or weapons, for example [6, 9, 10]. This view primarily originated from religious concepts and deontological considerations. After reviewing the taxonomy of sustainable investments of the EU's High-Level Expert Group and the definitions of the European Banking Authority, the Deutsche Bundesbank, lobbyists and private providers, we have not found a single indication that an investment's monetary performance should even partially determine its ethicality or sustainability. Despite the differences among ethical concepts in investment, certain investment contexts are touted as more unethical than others and the profitability of investment decisions are not used as criterion for distinction.

This paper examines the behavioral relationship between ethical investments and monetary success. Our research question examines whether and to what extent the economic outcome of an investment decision matters empirically for laypeople's moral evaluation of the investment decisions.

The remainder of this article unfolds as follows. First, we describe philosophical literature on moral luck and behavioral literature on outcome bias. Second, we report the design and results of our experimental research. Third, we discuss the implications of our results and identify the limitations of our study.

## The concept of moral luck

When is an action judged as moral or immoral? This question depends strongly on different ethical approaches. Kant [11] argues that a moral evaluation's most important criterion is the good will of the actor, which originates from an individual striving to fulfill concrete duties. These duties originate in moral reasoning and, in Kant's [11] case, with categorical imperative.

> "It is impossible to think of anything at all in the world, or indeed even beyond it, that could be considered good without limitation except a good will." [11, p. 7 (4:393)]

Furthermore, Kant [11] has pointed out that neither good luck nor bad luck should have any influence on our moral judgements about people or actions. Following this approach, consequences should not be taken into account when making moral judgments, particularly ones that cannot be foreseen by the acting individual [11, p. 8 (4:394)].

Williams [12] and Nagel [13] have confronted Kant's position in their well-known, identically titled essays, "Moral Luck." Williams [12] argues that the goal to make morality independent of luck is doomed to fail because our moral judgment clearly depends on the outcome. To illustrate his argument, Williams [12] uses the example of the French artist Gauguin, who has chosen to live as an artist in Tahiti rather than a life with a family. He argues that the moral judgment of this decision will depend on whether Gauguin becomes a successful artist or not [12].

A similar example of moral evaluation includes drunk driving. A drunk driver injuring a person on the way home from a party would lead to a different moral judgment than if the drunk driver arrives home without incident. The fact that both scenarios induce moral condemnation is not the point here. What is important is that the scenario without an incident is likely to receive less condemnation than the scenario with an incident. Because the car is driven negligently in both scenarios, this moral judgment should be identical, but Williams [12] argues that this is not the case and uses the term "moral luck" for this puzzle.

Moral luck refers to cases where an agent is considered as an object of moral judgment, although the essential aspects of their judgment depend on factors beyond their control [2]. In his work, Nagel [13] developed these ideas further and distinguished between different types of luck: resultant, circumstantial, constitutive or causal. Resultant luck refers to decisions that are made with uncertainty, which can produce different results. This view of luck is the underlying approach to this article. In the empirical studies in this article, we consider investment decisions that are, by definition, made uncertainly because no probability calculation for a risk quantification can be made on the decision's outcome [14]. Michaelson [15] points out that moral luck is particularly relevant in a business context because many business decisions cannot be reversed but often have an impact on many people who were not initially involved.

The concept of moral luck is related to the behavioral phenomenon of outcome bias [3, 16, 17], which occurs when the evaluation of actions is influenced by factors that are not logically

justifiable. This bias was first described by Baron and Hershey [17]. Tversky and Kahneman [18] explain that human intuition and behavior is biased in many ways and that humans tend to use short-sighted heuristics and "rules of thumb" to make decisions. These biases can also be found in moral judgments [3]. Many studies have shown that people unconsciously act immorally, even against their own ethical standards [19]. Although ethically irrelevant factors do factually influence our moral decisions, the moral implications of actions can also have an effect on epistemological questions, despite being deemed irrelevant a priori. Most prominently, Knobe [20] has shown that the moral quality of an action's side effect transforms the intentionality that we ascribe to the actor.

Building on the theoretical considerations outlined above, we will examine a potential driver behind the moral evaluation of investment decisions. Does the arbitrary monetary outcome of an uncertain investment decision factually determine its moral evaluation? Does the outcome change how the nature of the decision is interpreted? Do people believe that a given investment decision should be declared as more moral if it has a positive monetary outcome and should lead to a reinterpretation of the decision itself? Nagel [13] believes that the luck of people under moral evaluation influences our judgment. In this article, we examine whether Nagel's [13] claim aligns with people's folk intuitions for investment decisions.

The concept of moral luck has already been examined in several empirical studies. For within-subjects design, Kneer and Machery [21] found that the majority rated lucky and unlucky outcomes as morally equivalent. However, for between-subjects design, moral evaluations were outcome driven. The result of their study demonstrated that the outcome effect is mainly induced by hindsight bias implying that people perceive past events as being more predictable than they actually were. In his study of moral luck, Cushman [22] focuses on the mental status of the evaluated individuals and found that the judgment of whether an act is considered moral is primarily based on the mental state of the agent, whereas the verdict of guilt and punishment is based on the mental state and the causal relationship between the agent and the harmful consequences. Young et al. [23] have shown that moral luck depends more on wrong beliefs than on negative results and that participants with false beliefs are regarded as having less justified beliefs and, therefore, as moral culprits. Olson et al. [24, 25] found that across multiple cultures, even three-year-old children prefer people who experience good luck to those who suffer from bad luck.

In contrast to existing literature, our study refers to the realm of investment decisions and attempts to shed light on the implications of increasing demand for ethical investments combined with a lack of a clear-cut definition of ethical investment. We not only focus on the moral evaluation of decisions, but also explicitly investigate whether the very nature of an investment decision is interpreted differently in light of concrete outcomes.

## Study 1: Do people base their evaluations and interpretations on outcomes?

### Aim and design of Study 1

We used a between-subjects design with vignettes to investigate whether people's moral evaluation and interpretation of an investment decision hinges on its outcome, even though this outcome is beyond the investor's control. In vignette studies, test persons are presented with a hypothetical situation for which they must then answer questions [26]. In our vignette, an investment decision was presented. We implemented a 2 x 3 design. In two outcome conditions, we varied whether the investment decision resulted in a profit or loss. In three context conditions, we varied whether the investor worked for a bank, an automotive company or a municipality. Subjects were randomly assigned to one of the six resulting treatments. In the

following, we only present the vignettes with a positive and negative outcome for the bank context. The emphasis is only added here for means of illustration to highlight the lone difference between the two vignettes. The complete set of vignettes can be found in the S3 Data.

The vignette for the profit case reads as follows.

"Amanda Roberts is the Director of Financial Planning and Analysis of a large bank. Mrs. Roberts is responsible for the financial planning of the bank and has the goal to increase the financial assets. Several years ago, after consulting with financial advisors, she decided to buy complex financial products whose price development depends on the fluctuations of the oil price. Due to a significant oil price change in the following years, the bank recently *recorded a large profit.*"

The vignette for the loss case reads as follows.

"Amanda Roberts is the Director of Financial Planning and Analysis of a large bank. Mrs. Roberts is responsible for the financial planning of the bank and has the goal to increase the financial assets. Several years ago, after consulting with financial advisors, she decided to buy complex financial products whose price development depends on the fluctuations of the oil price. Due to a significant oil price change in the following years, the bank recently *had to accept a strong loss.*"

After the described situation, the test persons were asked two questions in randomized order to exclude series effects. One question asked for a moral evaluation of the investment decision. Another question asked them to label the investment decision to assess people's interpretation of the decision. To assess the investment morally, we asked "How would you evaluate Mrs. Roberts's behavior?" Respondents could then place a slider on a selection bar where the left pole meant "very moral" and the right pole meant "very immoral." For our analysis, we translate this choice into an "immorality score," (i.e., an integer from 0 [very moral] to 100 [very immoral]). To assess respondents' interpretation of the nature of the investment decision, we asked "What would you call Mrs. Roberts's behavior?" The possible answers were "Mrs. Roberts was investing," "Mrs. Roberts was speculating" and "Mrs. Roberts was gambling." Finally, we asked some demographic questions.

## Results of Study 1

Respondents were recruited via the CloudResearch Prime Panel [27]. All subjects were US inhabitants and at least 18 years old. No further restrictions were made. The survey was conducted in April 2020. The sample consisted of 367 subjects, resulting in at least 50 subjects per group. Based on data from a pilot, we had conducted a power analysis to determine the appropriate sample size for our study. We calculated with an expected mean difference of 10 points of the immorality score between the three contexts in the profit case and the three contexts in the loss case and a standard deviation of 24. Table 1 describes the subjects' age (life years) and

**Table 1. Age and gender by outcome in Study 1.**

|                       | overall        | profit        | loss          | profit vs. loss |
|-----------------------|----------------|---------------|---------------|-----------------|
| **n**                 | 367            | 169           | 198           |                 |
| **age (years)**       | 50.62 (15.53)  | 50.47 (15.33) | 50.75 (15.74) | p = 0.867       |
| **proportion of females** | 0.59 (9.42) | 0.60 (6.37)   | 0.58 (6.94)   | p = 0.672       |

Note: Reported are means and, in parentheses, standard deviations; p-value for age is based on unpaired t-test; p-value for gender is based on the chi-squared test.

**Table 2. Immorality scores and proportions using label "investment" by outcome.**

|  | overall | profit | loss |
|---|---|---|---|
| **n** | 367 | 169 | 198 |
| **immorality score (0 = very moral, 100 = very immoral)** | 34.42 (23.18) | 31.08 (23.45) | 37.27 (22.62) |
| **proportion using label "investment"** | 0.39 (9.34) | 0.45 (6.47) | 0.33 (6.62) |

Note: Reported are means and, in parentheses, standard deviations.

gender (proportion of females) overall and in the profit and loss condition of Study 1. A comparison between the demographic variables in the profit and loss condition indicates that their random assignment to these conditions was effective. Table 2 summarizes the descriptive statistics per outcome condition for Study 1.

Fig 1 illustrates respondents' "immorality score" (that could take an integer from 0 to 100) by outcome for the three investment contexts. Across all investment contexts, the average "immorality score" increased from 31.08 (sd = 23.45) in case of a profit to 37.27 (sd = 22.62) in case of a loss. To analyze whether the investment's outcome predicted the investment decision's moral evaluation, we first performed a one-way ANOVA with the "immorality score" as the dependent variable and the outcome of the investment decision (profit or loss) as the independent variable. We found that the outcome of the investment decision significantly predicted its moral evaluation (see Table 3). The same result is obtained when a Mann-Whitney U test is conducted (W = 19,477, p = 0.007).

To check for the robustness of this finding, we calculated a multi-way ANOVA that also includes the order in which questions were presented (moral assessment first, labeling first),

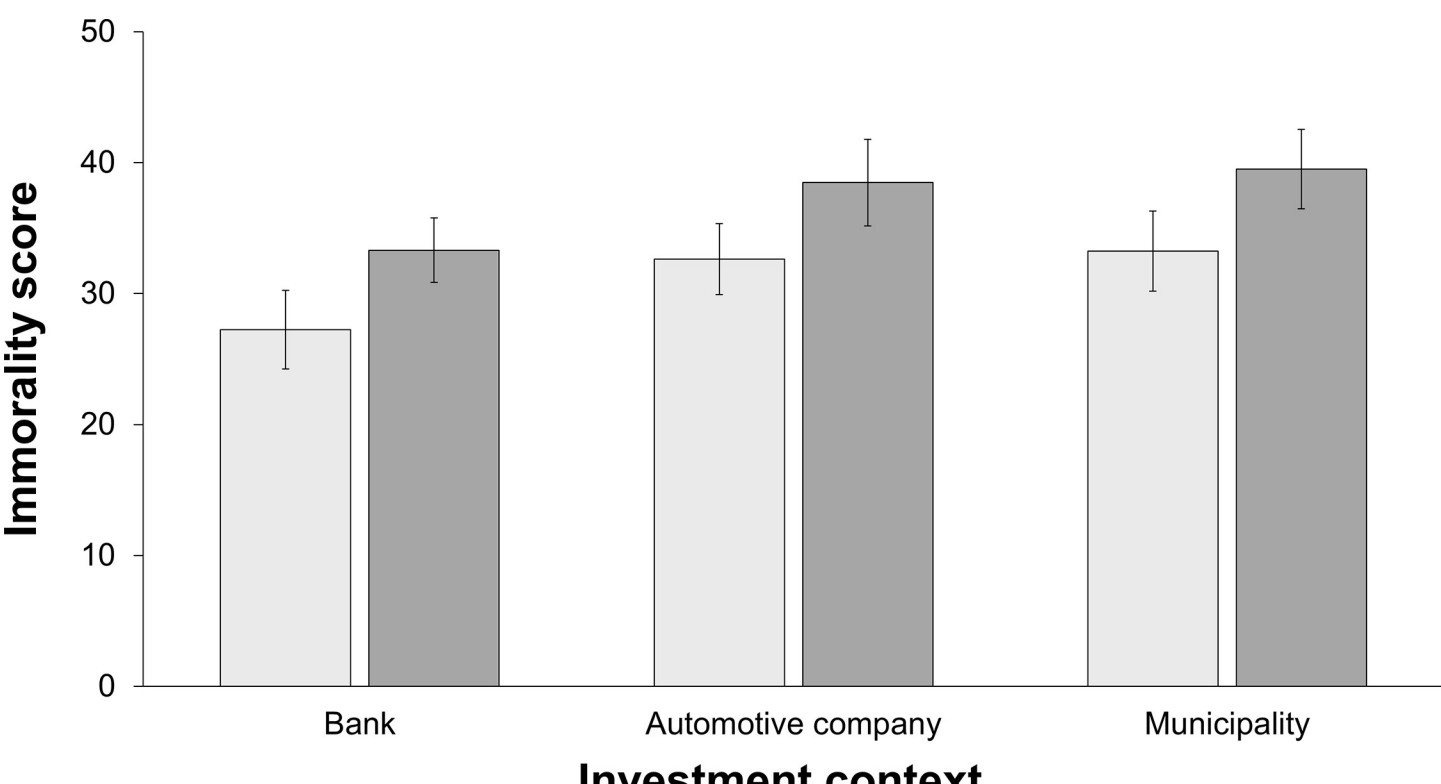

**Fig 1. Immorality score by outcome.** Note: The light-gray bars depict the profit case, and the dark-gray bars depict the loss case.

**Table 3. One-way analysis of variance using immorality score as the criterion.**

|  | Df | Sum Sq | Mean Sq | F | p |
|---|---|---|---|---|---|
| **outcome** | 1 | 3,488 | 3,488 | 6.59 | **0.011** |
| **residuals** | 365 | 193,176 | 529 |  |  |

the investment context (bank, company, municipality) and the interactions between both these factors and the outcome as the independent variables. Table 4 summarizes the results of the multi-way ANOVA. We found again that the outcome of the investment decision significantly predicted its moral evaluation, whereas other factors and their interactions with the outcome did not significantly influence the moral evaluation of the investment.

Fig 2 illustrates respondents' interpretation of the nature of the described investment decision by outcome for the three investment contexts. Across all investment contexts, the proportion of people labeling the decision as "investment" instead of the more pejorative terms "speculation" or "gambling" decreased from 44.97% (76 of 169 respondents) in case of a profit to 33.33% (66 of 198 respondents) in case of a loss.

To analyze whether the investment decision's outcome predicted respondents' likelihood to label the decision "investment" as opposed to "speculation" or "gambling," we calculated a logistic regression with a dummy variable for the label (1 = "investment," 0 = otherwise) as the dependent variable. Our independent variables were again the outcome of the investment decision, the order in which questions were presented, the investment context and the interactions between both of these factors and the outcome. As Table 5 summarizes, the likelihood of the decision being labeled as investment drops significantly if the investment happens to result in a loss as opposed to a profit.

However, the interaction effect between the investment's outcome and the dummy variable of whether respondents labeled the decision before or after evaluating is highly significant. Table 6 therefore summarizes the results of a logistic regression which only considers the labeling of those respondents who labeled the decision before they evaluated it. In this case, the outcome of the investment no longer predicts its labeling. It thus seems that reflecting on the investment's morality influences respondents' interpretation of the investment decision.

## Study 2: Does the bias against unprofitable investments persist upon reflection?

### Aim and design of Study 2

The aim of Study 2 was to analyze whether the tendency observed in Study 1 to base the moral evaluation and the interpretation of the nature of the investment decision on outcomes is grounded in moral reasoning or represents an unreflected moral intuition. Put differently, we want to understand whether the observed discrimination against unlucky investors is

**Table 4. Multi-way analysis of variance using immorality score as the criterion.**

|  | Df | Sum Sq | Mean Sq | F | p |
|---|---|---|---|---|---|
| **outcome** | 1 | 3,488 | 3,488 | 6.568 | **0.011** |
| **order** | 1 | 3 | 3 | 0.006 | 0.939 |
| **context** | 2 | 2,503 | 1,252 | 2.357 | 0.096 |
| **outcome × order** | 1 | 30 | 30 | 0.056 | 0.813 |
| **outcome × context** | 2 | 1 | 0 | 0.001 | 0.999 |
| **residuals** | 359 | 190,639 | 531 |  |  |

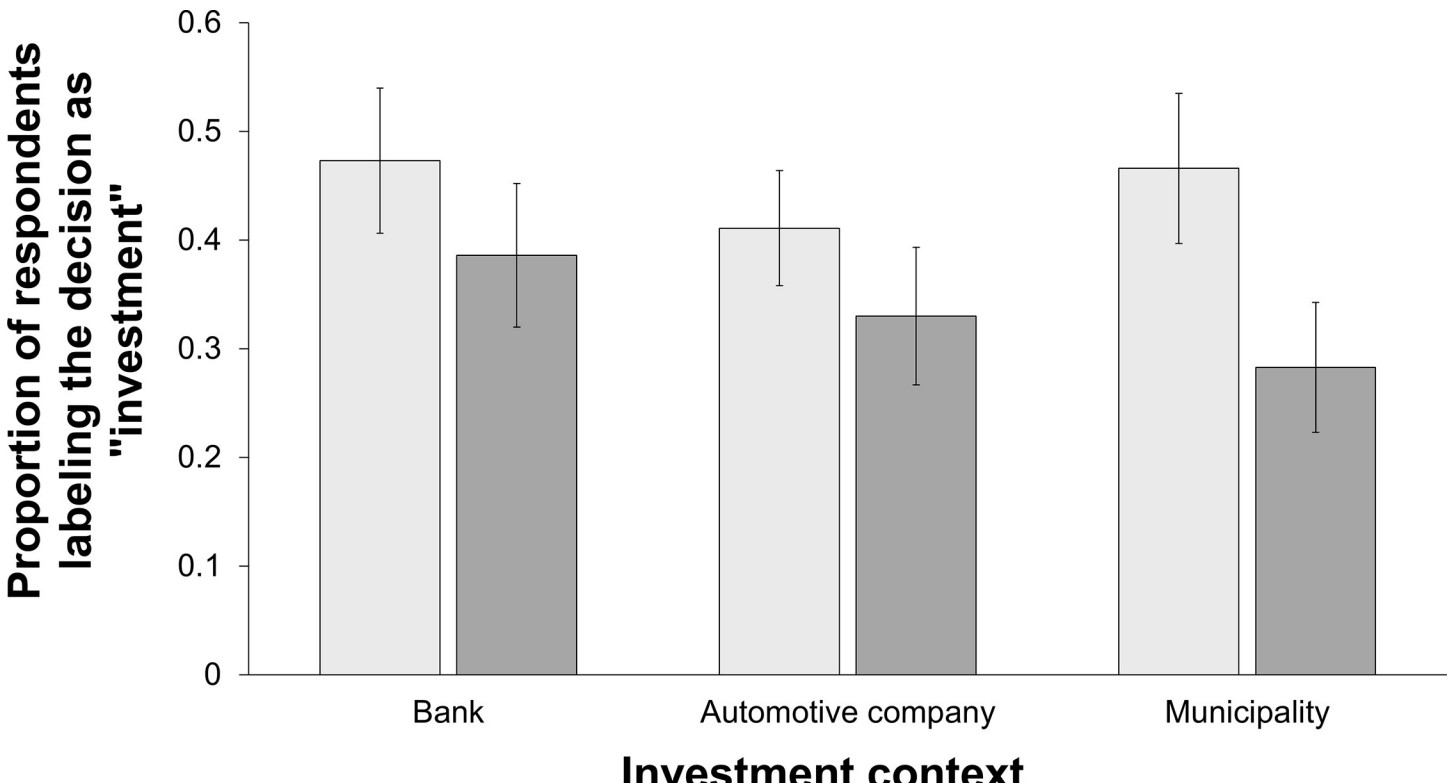

**Fig 2. Proportion of respondents labeling the decision "investment" by outcome.** Note: The light gray bars depict the profit case and the dark gray bars depict the loss case.

conscious or unconscious bias. The design of Study 2 closely followed the design of Study 1. The exception was that we let respondents evaluate and label the investment decision before the outcome unfolded (i.e., right after the decision had been taken). Only on the next screen did we inform respondents that it turned out in the following years that the investment decision had led to a loss or to a profit depending on the treatment. They were then explicitly asked to reconsider their previous evaluation in light of this new information. If they wanted to change their initial evaluation of the investment decision after reconsidering it, they were free to do so. Otherwise, they could simply adopt their initial evaluation. The same reconsideration took place for the labeling of the decision. As in Study 1, we randomized whether respondents were first asked for their moral evaluation or for their interpretation. The order in which

**Table 5. Logistic regression with a dummy for label as the dependent variable.**

|  | Estimate | Std. Error | z value | p |
|---|---|---|---|---|
| **(intercept)** | 0.092 | 0.323 | 0.287 | 0.774 |
| **outcome (0 = profit, 1 = loss)** | -0.979 | 0.461 | -2.123 | **0.034** |
| **order (0 = eval first, 1 = label first)** | -0.359 | 0.312 | -1.151 | 0.250 |
| **automotive** | -0.283 | 0.386 | -0.734 | 0.463 |
| **municipality** | -0.046 | 0.379 | -0.121 | 0.904 |
| **outcome × order** | 1.179 | 0.455 | 2.595 | **0.009** |
| **outcome × automotive** | -0.145 | 0.533 | -0.271 | 0.786 |
| **outcome × municipality** | -0.404 | 0.563 | -0.719 | 0.472 |

**Table 6. Logistic regression with a dummy for label as the dependent variable for label first.**

|  | Estimate | Std. Error | z value | P |
|---|---|---|---|---|
| (intercept) | -0.325 | 0.364 | -0.894 | 0.371 |
| outcome (0 = profit, 1 = loss) | 0.325 | 0.525 | 0.620 | 0.535 |
| automotive | -0.205 | 0.540 | -0.380 | 0.704 |
| municipality | 0.057 | 0.518 | 0.110 | 0.912 |
| outcome × automotive | -0.254 | 0.709 | -0.359 | 0.720 |
| outcome × municipality | -0.750 | 0.774 | -0.970 | 0.332 |

respondents reconsidered their answers after having received the information on the outcome was identical to the order in which they made their initial assessments.

## Results of Study 2

Respondents were also recruited via the CloudResearch Prime Panel. All subjects were again US inhabitants and at least 18 years old. No further restrictions applied. The survey was also conducted in April 2020. Subjects who had participated in Study 1 were excluded from the sample. The sample consisted of 368 subjects, resulting again in at least 50 subjects per group. Table 7 describes the subjects' age (life years) and gender (proportion of females) overall and in the profit and loss condition of Study 2. A comparison between the demographic variables in the profit and loss condition indicates that their random assignment to these conditions was effective. Table 8 summarizes the descriptive statistics per outcome condition for Study 2.

To understand whether subjects wanted to revise their initial evaluation, we check whether their reconsidered immorality scores differed from their initial ones. The difference between initial evaluations in case of a profit and in case of a loss was not statistically significant (36.64 vs. 32.49, p = 0.259, Mann-Whitney U test). This is captured by a "change score" that is measured as the difference between a given subject's reconsidered immorality score and their initial immorality score. Across all investment contexts, the average "change score" increased from 0.59 (sd = 18.76) in case of a profit to 5.43 (sd = 17.49) in case of a loss. This result is illustrated in Fig 3.

To analyze whether the quality of the investment's outcome predicted the inclination to revise an investment's moral evaluation in light of the outcome, we first performed a one-way ANOVA with the "change score" as the dependent variable and the outcome of the investment decision (profit or loss) as the independent variable. The result is that the outcome of the investment decision predicted the magnitude of the revision as measured by the change score (see Table 9).

As in Study 1, we performed a robustness test by calculating a multi-way ANOVA, which also includes the order in which questions were presented (moral assessment first, labeling first), the investment context (bank, company, municipality) and the interactions between both of these factors and the outcome as independent variables. Table 10 summarizes the

**Table 7. Age and gender by outcome in Study 2.**

|  | overall | profit | loss | profit vs. loss |
|---|---|---|---|---|
| n | 368 | 188 | 180 |  |
| age (years) | 45.54 (16.30) | 45.40 (16.96) | 45.68 (15.64) | p = 0.868 |
| proportion of females | 0.64 (9.21) | 0.65 (6.54) | 0.63 (6.48) | p = 0.753 |

Note: Reported are means and, in parentheses, standard deviations; p-value for age is based on unpaired t-test; p-value for gender is based on chi-squared test.

**Table 8. Change scores and proportions relabeling decision by outcome.**

|  | overall | profit | loss |
|---|---|---|---|
| **n** | 368 | 188 | 180 |
| **change score** | 2.95 (18.29) | 0.59 (18.76) | 5.43 (17.49) |
| **proportion relabeling decision as "investment"** | 0.033 (3.43) | 0.048 (2.93) | 0.017 (1.73) |
| **proportion relabeling decision as "speculation" or "gambling"** | 0.068 (4.83) | 0.027 (2.22) | 0.111 (4.21) |

Note: Reported are means and, in parentheses, standard deviations

results of the multi-way ANOVA and confirms the effect of the investment's outcome on the change score (i.e., the tendency to reevaluate the immorality of the investment).

Fig 4 illustrates respondents' tendency to reinterpret a decision after getting to know its outcome. The difference between the initial proportions of subjects who labeled a decision as investment in case of a profit and in case of a loss was not statistically significant (53.7% vs. 57.8%, p = 0.434, chi-squared test). Across all investment contexts, the proportion of people who labeled the decision "investment" ex ante to relabel it "speculation" or "gambling" ex post was 2.66% (5 of 188) in case of a profit and 11.11% (20 of 180) in case of a loss. Vice versa, the proportion of people who labeled the decision "speculation" or "gambling" ex ante to relabel it "investment" ex post was 4.79% (9 of 188) in case of a profit and 1.67% (3 of 180) in case of a loss.

To analyze whether the investment decision's outcome predicted respondents' likelihood to relabel the decision, we calculated a logistic regression with a dummy variable for the label change (0 = no, 1 = yes) as the dependent variable. Our independent variables were the outcome of the investment decision, the ex ante labeling of the decision, the order in which questions were presented, the investment context, and the interactions of the investment's outcome with the other variables. As Table 11 summarizes, there is a significant interaction effect between the investment's outcome and the ex ante labeling of the decision on the propensity to relabel the decision.

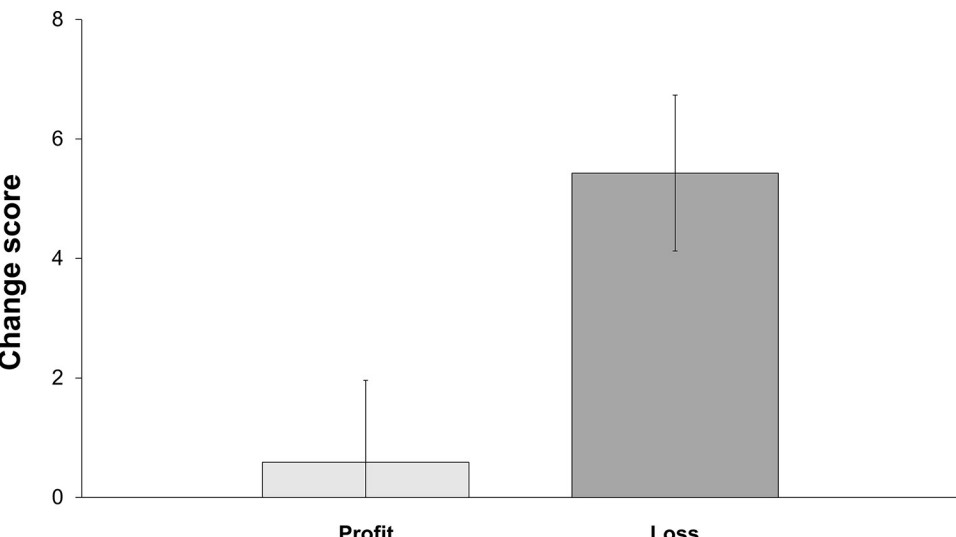

**Fig 3. Change score by outcome.** Note: The change score is the difference between reconsidered and initial immorality scores. The light gray bar depicts the profit case and the dark gray bar depicts the loss case.

**Table 9. One-way analysis of variance using change score as the criterion.**

|  | Df | Sum Sq | Mean Sq | F | p |
|---|---|---|---|---|---|
| outcome | 1 | 2,157 | 2156.5 | 6.545 | **0.011** |
| residuals | 366 | 120,600 | 329.5 |  |  |

**Table 10. Analysis of variance using change score as the criterion.**

|  | Df | Sum Sq | Mean Sq | F | p |
|---|---|---|---|---|---|
| outcome | 1 | 2,157 | 2,157 | 6.589 | **0.011** |
| order | 1 | 198 | 198 | 0.604 | 0.438 |
| context | 2 | 492 | 246 | 0.751 | 0.473 |
| outcome × order | 1 | 675 | 675 | 2.063 | 0.152 |
| outcome × context | 2 | 1,405 | 702 | 2.146 | 0.118 |
| residuals | 360 | 117,830 | 327 |  |  |

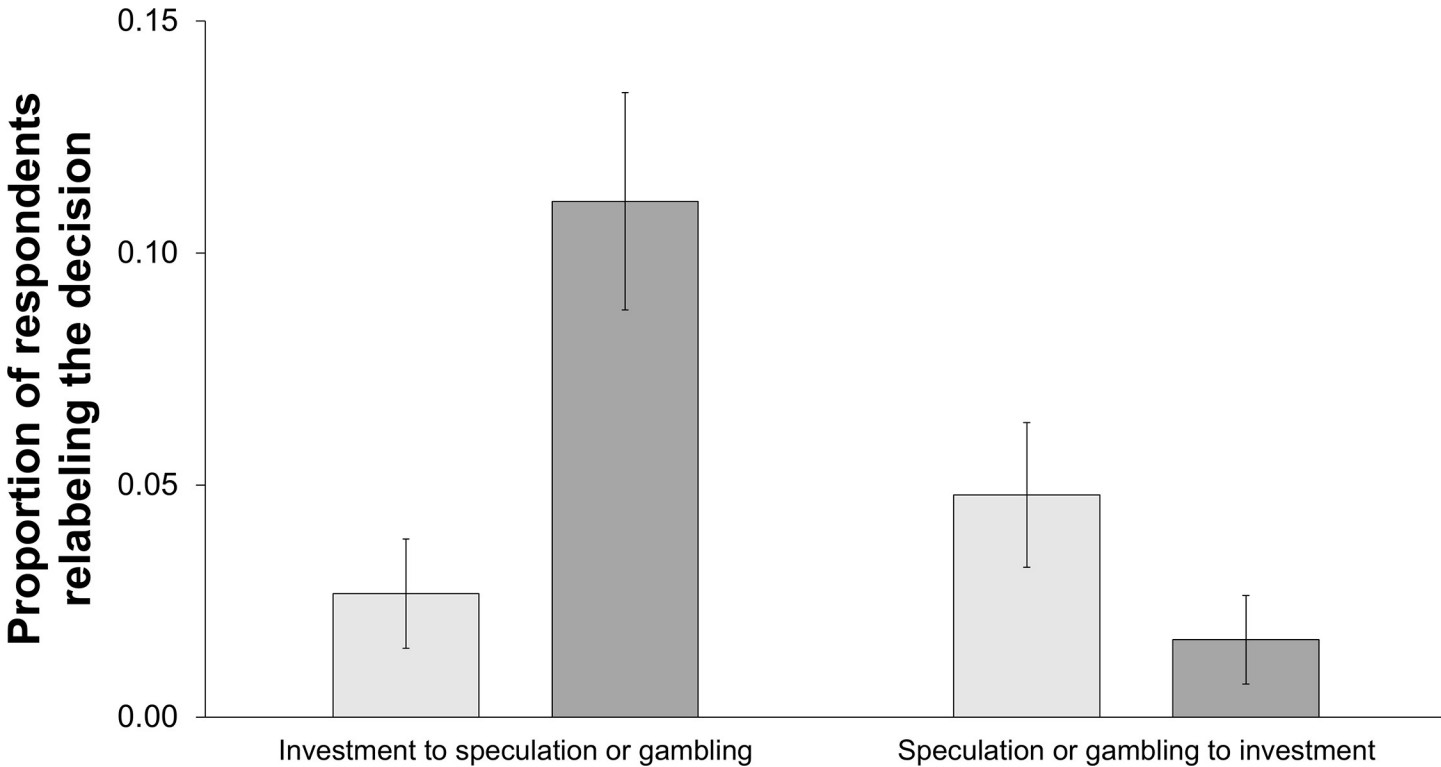

**Fig 4. Proportion of respondents relabeling the decision by outcome.** Note: The light-gray bar depicts the profit case and the dark-gray bar depicts the loss case.

## Implications and limitations

The results of our study should be interpreted in the context of the ethical discourse with respect to credit rating agencies (CRAs), the valuation of issues and the taxonomy debate of the EU on sustainable finance. The assignment of ratings by CRA has a strong impact on lenders and borrowers. In addition to the documented conflict of interest in the valuation of bond and creditor issues that is based on the payment links and the lack of transparency [28, 29], we

**Table 11. Logistic regression with a dummy for label change as dependent variable.**

| | Estimate | Std. Error | z value | p |
|---|---|---|---|---|
| **(intercept)** | -2.397 | 0.618 | -3.878 | **0.000** |
| **outcome (0 = profit, 1 = loss)** | -0.516 | 0.918 | -0.562 | 0.574 |
| **ex_ante_label (1 = investment, 0 = other)** | -0.844 | 0.583 | -1.450 | 0.147 |
| **order (0 = eval first, 1 = label first)** | 0.072 | 0.560 | 0.129 | 0.897 |
| **automotive** | 0.562 | 0.679 | 0.828 | 0.408 |
| **municipality** | 0.047 | 0.734 | 0.064 | 0.949 |
| **outcome × ex_ante_label** | 2.686 | 0.870 | 3.086 | **0.002** |
| **outcome × order** | 0.164 | 0.729 | 0.226 | 0.822 |
| **outcome × automotive** | -1.183 | 0.875 | -1.352 | 0.176 |
| **outcome × municipality** | -0.908 | 0.932 | -0.974 | 0.330 |

point out another potential conflict based on the results of our study. Sustainability rating companies that are commissioned to issue a Second Party Opinion (SPO) for so-called green or social bonds should make the valuation of a bond or debenture independent of its issuer. This is because the issuer will have a more or less successful history with regard to return on investment that might bias the moral evaluation of the issued bonds. Assuming our respondents' inclination to evaluate profitable investments as more moral and to do so even upon reflection should sensitize us for the strength of the outcome bias in evaluating investments.

Because the rating of financial instruments as ethical constitutes a public good, we should strive to improve the procedures from which they result as much as possible [30]. For an ethical evaluation of investments to be valuable, it should be orthogonal to the evaluation of financial performance. The assessment of whether an investment is ethical or socially responsible should therefore be based on a comprehensible and transparent catalogue that provides an unambiguous basis for its ex ante evaluation. The enterprise of adding an ethical dimension to the assessment of investments becomes at best empty if it is highly correlated with the investment's financial success.

Our study is subject to some limitations. First of all, the identified outcome bias may be mitigated when considering the choices of experts in the field of ESG. Moreover, the "CloudResearch Prime Panel" used in our study is not representative of the overall US population. Specifically, the panel has a higher education, is younger, is more liberal and is less religious than the US population [31]. Krische [32] also points out that the level of financial literacy in this sample is higher than in the average US population. Further studies with a more diverse sample are needed to gain an understanding for the correlation of personal characteristics and the moral evaluation and interpretation of investment decisions. Finally, it should be noticed that no main effect of outcome on label change could be identified in Study 2. This does not provide any evidence for the absence of this effect but implies that no evidence for its presence was identified.

## Conclusion

Literature in the sustainable investment discourses has dedicated much ink for outlining and elaborating on the meaning of ethical investments. Doing so, the discourse has largely concentrated on the underlying asset of investment decisions. Although the focus on the underlying asset affects public perception, literature on moral luck suggests that moral perceptions also depend on the success of the action, which means that successful actions are perceived as morally superior to unsuccessful ones. In our paper, we tested whether moral luck matters empirically for the moral judgment of investment decisions as well, although this is not proposed by

any of the numerous conceptualizations of ethical investments in the normative literature. Specifically, we measured the extent to which the moral evaluation and interpretation of the nature of investment decisions depended on the investment's arbitrary monetary outcome.

We found that investment decisions that turned out to be less profitable tended to be evaluated more negatively and were also more often interpreted as speculation or gambling instead of investments. Notably, our participants were even downgrading the morality of an investment and relabeled it as "speculation" or "gambling" more often when having the hindsight of a bad outcome. This divergence between ex-ante and ex-post evaluation of investments shows that the outcome bias observed in Study 1 is conscious rather than unconscious. The reason for this is that it is based on an intentional action and does not merely emerge as a subtle between-subjects effect. Notably, this is in contrast to the results obtained by Bazerman and Tenbrunsel [3] in the context of scientific fraud cited at the beginning of this article.

The results of our studies lead us to the conclusion that a rigorous, normative definition of what constitutes an ethical investment decision is indispensable to mitigate the influence of behavioral biases in the ethical evaluation of investments as the one exemplified in this study. When the leeway in classifying investments against the background of a vague definition is greater, the potential influence of the behavioral biases of the classifier is stronger. In any case, it seems likely from our findings that investments labeled as ethical due to the nature of the underlying asset might nonetheless face stronger public rejection if resulting in a financial loss. In this context, our results also suggest that experts should be wary of how they describe investment decisions, as the terms "investment" and "speculation" that may sometimes be used interchangeably in expert jargon do actually come with different moral connotations for laypeople. Clear definitions and generally applicable criteria for the moral valuation of funds remain a major challenge of utmost importance because the subjective classification of funds as ethical or sustainable may be subject to behavioral factors that could be difficult to defend on normative grounds.

## Supporting information

**S1 Data. Data of Study 1.**
(XLSX)

**S2 Data. Data of Study 2.**
(XLSX)

**S3 Data. Vignettes.**
(DOCX)

## Acknowledgments

We are grateful for the very constructive comments of three anonymous reviewers.

## Author Contributions

**Conceptualization:** Raphael Max, Matthias Uhl.

**Data curation:** Raphael Max, Matthias Uhl.

**Formal analysis:** Raphael Max, Matthias Uhl.

**Funding acquisition:** Raphael Max, Matthias Uhl.

**Investigation:** Raphael Max, Matthias Uhl.

**Methodology:** Raphael Max, Matthias Uhl.

**Project administration:** Raphael Max, Matthias Uhl.

**Resources:** Raphael Max, Matthias Uhl.

**Software:** Raphael Max, Matthias Uhl.

**Supervision:** Raphael Max, Matthias Uhl.

**Validation:** Raphael Max, Matthias Uhl.

**Visualization:** Raphael Max, Matthias Uhl.

**Writing – original draft:** Raphael Max, Matthias Uhl.

**Writing – review & editing:** Raphael Max, Matthias Uhl.

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
