## [Decision Letter · Decision Letter 0]

25 Mar 2022

PONE-D-22-02964Morally Lucky Investors: We find profitable investments more moral although we think we should notPLOS ONE

Dear Dr. Max,

Thank you for submitting your manuscript to PLOS ONE. After careful consideration, we feel that it has merit but does not fully meet PLOS ONE’s publication criteria as it currently stands. Therefore, we invite you to submit a revised version of the manuscript that addresses the points raised during the review process.

We have received three reports of experts in the topic, and I agree with them that the paper is interesting but it needs substantial changes. In particular, the theoretical discussion on the concept of moral luck needs to be adjusted. Moreover, the statistical analysis is not adequate in some points: the use of ANOVA or other tests need to be justified, specially when it is not well suited, as in the case of binary variables. Please, read carefully the recommendations of the reviewers and try to update your analysis following their comments.

We look forward to receiving your revised manuscript.

Kind regards,

Alfonso Rosa Garcia

Academic Editor

PLOS ONE

Journal Requirements:

Reviewers' comments:

Reviewer's Responses to Questions

**Comments to the Author**

1. Is the manuscript technically sound, and do the data support the conclusions?

Reviewer #1: No

Reviewer #2: Partly

Reviewer #3: Partly

2. Has the statistical analysis been performed appropriately and rigorously? 

Reviewer #1: I Don't Know

Reviewer #2: No

Reviewer #3: Yes

3. Have the authors made all data underlying the findings in their manuscript fully available?

Reviewer #1: No

Reviewer #2: Yes

Reviewer #3: Yes

4. Is the manuscript presented in an intelligible fashion and written in standard English?

Reviewer #1: Yes

Reviewer #2: Yes

Reviewer #3: Yes

5. Review Comments to the Author

Reviewer #1: The paper explores an interesting hypothesis but seems to over-state the empirical support for it. It would also benefit from further conceptual clarification.

Major comments

⁃ p.5: clarify the conception of moral luck. A good action is not equivalent to a good will. The Kant quote does not say “that the most important criterion of moral evaluation is the good will”

⁃ from p.11: provide more details on both studies: data, details, and results of analyses such as a table. Explain why you used this analysis rather than a one-way ANOVA or Wilcoxon test, or report results of these analyses. Provide reasons for the chosen sample sizes for both studies.

⁃ Relatedly: Study 2, p.13: Please do a sanity check that there is no statistically detectable difference in initial evaluations and labels between groups. Test for differences in immorality scores and labels before versus after outcome.

⁃ Discuss the null results of your studies in the limitations

Minor comments

⁃ p.3: Williams has coined “moral luck” as an oxymoron: provide exact reference, or remove this claim (did he really call it an “oxymoron”? Would be news to me)

⁃ p.3: ditto for “Nelkin argues that immunity from luck is seen by many [who?] as the very essence of morality”

⁃ p.4, second para should be clearer; research question should be stated succinctly

⁃ p.4: historical note: I would bet that there have been ethical restrictions on investments long before the 20th century. Consider Christian restrictions on interest in the Middle Ages or “halal” investment in Muslim Law.

⁃ p.4: definitions of “ethical investment” need further clarification. For one thing, clarify “ethical”. You seem to suggest that “non-financial” is identical with “ethical”, which is incorrect. Also, please explain why investment in alcohol or tobacco should be unethical.

⁃ p.7: explain “Knightian uncertainty”

⁃ p.7: sentence unclear, please revise: “While ethically irrelevant factors exercise a factual influence on our moral decisions, moral implications…”

⁃ p.8: “our intention would reject this”: unclear, revise

⁃ p.15: “it would be better if an institutionalised mechanism controlled that such evaluations took place behind a veil of ignorance”: either explain in detail how this follows from your studies, or remove

Reviewer #2: Review of Manuscript # PONE-D-22-02964

Title: Morally Lucky Investors: We find profitable investments more moral although we think we should not

Summary: This paper reports the results of two experiments conducted to examine whether individuals evaluate the morality of investment decisions through a lens of moral luck.

In the first experiment, participants learn that an investment decision was made and that the investment was later determined to be either profitable or unprofitable (manipulated between subjects). Participants evaluate the profitable investment as more moral than the unprofitable investment and were more likely to label the profitable investment as an “investment” as opposed to “speculation” or “gambling.”

In the second experiment, participants learn about the investment decision and then make assessments of the morality of the investment and the appropriate label before learning whether the investment was profitable or not. Participants then learn about the investment’s outcome (profit or loss) and can revise their judgments on morality and labeling. Results of this study show no differences in morality judgments or labeling between the profit and loss condition, suggesting that individuals know that they should not base these judgments on the outcome of the investment decision.

Comments on Methods:

1.) Participants: The authors should tabulate or otherwise describe the demographics of the participants in the studies and indicate whether random assignment of participants to conditions was effective. That is, are there differences in demographics across experimental conditions? If there are, the authors should statistically control for demographic variables that differ.

2.) Descriptive Statistics: The authors should tabulate full descriptive statistics (means, standard deviations, number of participants, etc.) by condition for the DVs. Currently, only the means/frequencies are shown in a bar graph format.

3.) Statistical Models:

a. The morality judgments in both experiments are formally tested using ANOVA. The authors should include tables for the estimated ANOVA models.

b. For the order of DV elicitation manipulation and the investment context manipulation, the authors should analyze the interaction of these factors with the primary IV of interest (whether the investment is profitable or not). The authors only report the main effects of these two factors on the DVs; however, the main effects are of less concern than the interactive effects when interpreting whether these factors impact the inferences drawn. Also, note that when adding the interaction terms to the ANOVA to account for the full-factorial design of the experiment, the error degrees of freedom will be reduced. I replicated the reported results for experiment 1 using the data provided and found that adding the interaction terms will not affect the inferences of that study.

c. The labeling judgments in both experiments are also tested using ANOVA. This approach may be inappropriate because the dependent variable is constructed as a binary indicator variable (1 = “investment” or 0 = other labels). This may lead to violation of certain assumptions of ANOVA. A more appropriate method would be logistic regression. Again, tables for the estimated models should be presented and the interaction terms should be analyzed.

d. In the second experiment results, it would be informative to know whether participants in any condition were more likely to change their morality and labeling judgments after learning of the investment outcome. This could be accomplished by calculating a change score (reconsidered morality judgment – initial morality judgment), or by using another method. At the least, it would be useful to know whether the initial judgments varied at all by condition.

Other Comments:

1. The study’s abstract only describes the first experiment, yet the study’s title and write-up contemplate both experiments. I recommend revising the abstract to reflect the inferences drawn from the second experiment. Likewise, the conclusion paragraph does not mention the inferences of experiment 2.

2. Footnote 1 cites German Criminal Code (§46 paragraph 2 StGB), but does not explain what that part of the Code requires. More explanation would be helpful for the reader.

3. The study is partially motivated through the idea that certain investment opportunities are often touted as being more ethical – investments with underlying focus on CSR, green initiatives, etc. However, there is a bit of disconnect between this motivation and what is operationalized in the study. The investment security in the study is described as a complex financial instrument with a value that fluctuates based on changes in oil prices. No mention is made of whether the underlying investment is geared toward an ethical cause. The authors could better describe this design choice in light of the overall goals of the research and how the research is motivated.

4. The authors describe investment decisions, generally, as decisions made under conditions of Knightian uncertainty. I believe this is too strong of a statement. Typically, we think of investors making (or at least trying to make) rational decisions based on some estimation of expected value of the investment – i.e., managing risk as opposed to true Knightian uncertainty. While some investment contexts will reflect this high level of uncertainty, not all will. This also disconnects with the operationalization of the context in the experiment. In the vignettes, the investor is described as having consulted with financial advisors, presumably as a rational investor weighing risk and potential reward.

5. In the conclusion, the inferences of the study are extrapolated to infer that sustainability rating professionals may succumb to the moral luck bias when providing opinions regarding the moral evaluation of particular investments; however, this may not be the case. In the experiments, non-professionals make judgments demonstrating the moral luck effect. It is not clear that these results would generalize to professionals dedicated to such evaluations.

Reviewer #3: Summary

In study 1, they found people rate higher immoral scores for a "loss" scenario than a "profit" scenario, similarly there are less people who believe it is an investment in the "loss" scenario" than the "profit scenario". Study 2 was to analyze whether the observed pattern in Study 1 is based on moral reasoning or an unreflected moral intuition.

Comments

1. The literature has already showed people exhibit outcome bias (related to moral luck). I think the authors really need to emphasize how their work contribute to the literature. For instance, for study 1, even without their study, at least some readers would already think that people would base their evaluations and interpretations on outcome.

2. The section on “the concept of moral luck” is a bit too long and nearly included an entire paragraph of the original words from Kant. I don’t think it is necessary. It would make the story to stay on track better.

3. Figure 2a appear to show the results after they see the outcome. I think readers would be interested to see how their ratings are immediately after they see only the decision but before the outcome is revealed.

4. Figure information can be better presented. For instance, the authors can easily include a legend name of what each color of the bar represent. The authors could also consider showing the p-values on the graph (or the show the ** on the graph).

Moreover, please provide a name for the figure, at the first glance Figure 1a and Figure 2a appear to be the same.

Overall, I would think it is an interesting exercise but can be explored in more depth.

6. PLOS authors have the option to publish the peer review history of their article (what does this mean?). If published, this will include your full peer review and any attached files.

Reviewer #1: No

Reviewer #2: No

Reviewer #3: No

---

## [Author Response · Author response to Decision Letter 0]

30 Jun 2022

Dear editor,

Thank you very much for allowing us to revise our manuscript and for the many helpful comments on our work.

Before we go into detail on all of the individual comments, we would like to briefly point out that a very helpful additional analysis suggested by Reviewer 2 has led to a change in the interpretation of our data. We have reanalyzed our data from scratch and now show that subjects' evaluation of the morality of an investment is indeed explicitly downgraded after the announcement of the economic success. We analyzed this and now discussed it in the paper. In line with a suggestion of Reviewer 2, we have labeled the respective measure "change score". Consequently, subjects not only labeled economically unsuccessful investment decisions as less moral, but explicitly changed the assessment after the outcome was revealed. This intentional divergence between ex-ante and ex-post evaluation of investments is in contrast to the results obtained by other scholars.

We have further tried to take all reviewer comments into account in our revision and will show you in detail below how we have dealt with reviewers’ comments. Reviewer comments are highlighted in bold type. Below that, you will find our comments in italics. We have created an individual PDF for all reviewers to address each comment individually for each reviewer.

We hope that our revision meets your expectations and are happy to address any additional comments you may have.

Your sincerely,

The authors

Reviewer #1:

Thank you very much for the constructive comments on our manuscript. We think that our manuscript has very much benefitted from incorporating the suggestions you made in your referee report. In what follows, we will address each of your comments individually.

⁃ p. 5: clarify the conception of moral luck. A good action is not equivalent to a good will. The Kant quote does not say “that the most important criterion of moral evaluation is the good will”

We specifically addressed this point and added more sentences to the manuscript.

⁃ from p. 11: provide more details on both studies: data, details, and results of analyses such as a table. Explain why you used this analysis rather than a one-way ANOVA or Wilcoxon test, or report results of these analyses. Provide reasons for the chosen sample sizes for both studies.

We added the data requested by Reviewer 1 to both studies.

⁃ Relatedly: Study 2, p. 13: Please do a sanity check that there is no statistically detectable difference in initial evaluations and labels between groups. Test for differences in immorality scores and labels before versus after outcome.

We now specifically investigate the differences and have transparently prepared the results and reported them in the article. As mentioned above, we included an analysis of the so called “change score” of the evaluations and analyzed the proportion of respondents’ relabeling the decision.

⁃ Discuss the null results of your studies in the limitations

The introduction of the change score and of the change in labels has led to an update in the results of Study 2 such that we do now observe different change scores and different inclinations to relabel the decision. The discussion of the results was changed accordingly.

⁃ p. 3: Williams has coined “moral luck” as an oxymoron: provide exact reference, or remove this claim (did he really call it an “oxymoron”? Would be news to me)

We have rechecked the citation "oxymoron" and provided the respective reference.

⁃ p. 3: ditto for “Nelkin argues that immunity from luck is seen by many [who?] as the very essence of morality”

Due to the fact that this is a direct quote, we would like to leave the sentence as it is. If the reviewer prefers to have the sentence removed, we will comply with this request.

⁃ p. 4, second para should be clearer; research question should be stated succinctly

We have included another paragraph to clarify and describe the research question.

⁃ p. 4: historical note: I would bet that there have been ethical restrictions on investments long before the 20th century. Consider Christian restrictions on interest in the Middle Ages or “halal” investment in Muslim Law.

The reviewer is right, of course, that there was a centuries-old restriction on investment in various religions. We had not originally negated this fact. We infer from the reviewer's comment that our original wording was misleading and have consequently now completed the paragraph to reflect the comment.

⁃ p. 4: definitions of “ethical investment” need further clarification. For one thing, clarify “ethical”. You seem to suggest that “non-financial” is identical with “ethical”, which is incorrect. Also, please explain why investment in alcohol or tobacco should be unethical.

We have also revised the wording here to meet the reviewer's objections.

⁃ p. 7: explain “Knightian uncertainty”

Since the term "Knightian uncertainty" is not essential for the continuation of the argumentation in the paper and the point would require further description, we have decided to dispense with the term at this point.

⁃ p. 7: sentence unclear, please revise: “While ethically irrelevant factors exercise a factual influence on our moral decisions, moral implications…”

We have adjusted the wording.

⁃ p. 8: “our intention would reject this”: unclear, revise

We have removed the sentence as it is not of great relevance for the further argumentation of the paper.

⁃ p. 15: “it would be better if an institutionalised mechanism controlled that such evaluations took place behind a veil of ignorance”: either explain in detail how this follows from your studies, or remove

We have removed the sentence as it is not of great relevance for the further argumentation of the paper.

 

Reviewer #2:

Thank you very much for the constructive comments on our manuscript. We think that our manuscript has very much benefitted from incorporating the suggestions you made in your referee report. In what follows, we will address each of your comments individually.

1.) Participants: The authors should tabulate or otherwise describe the demographics of the participants in the studies and indicate whether random assignment of participants to conditions was effective. That is, are there differences in demographics across experimental conditions? If there are, the authors should statistically control for demographic variables that differ.

We have added this information to the paper.

2.) Descriptive Statistics: The authors should tabulate full descriptive statistics (means, standard deviations, number of participants, etc.) by condition for the DVs. Currently, only the means/frequencies are shown in a bar graph format.

We have added this information to the paper.

a. The immorality judgments in both experiments are formally tested using ANOVA. The authors should include tables for the estimated ANOVA models.

We have added this information to the paper.

b. For the order of DV elicitation manipulation and the investment context manipulation, the authors should analyze the interaction of these factors with the primary IV of interest (whether the investment is profitable or not). The authors only report the main effects of these two factors on the DVs; however, the main effects are of less concern than the interactive effects when interpreting whether these factors impact the inferences drawn. Also, note that when adding the interaction terms to the ANOVA to account for the full-factorial design of the experiment, the error degrees of freedom will be reduced. I replicated the reported results for experiment 1 using the data provided and found that adding the interaction terms will not affect the inferences of that study.

 We have adapted the analysis according to this suggestion.

c. The labeling judgments in both experiments are also tested using ANOVA. This approach may be inappropriate because the dependent variable is constructed as a binary indicator variable (1 = “investment” or 0 = other labels). This may lead to violation of certain assumptions of ANOVA. A more appropriate method would be logistic regression. Again, tables for the estimated models should be presented and the interaction terms should be analyzed.

We are grateful for this indication and are now using a logistic regression in the revised version of our paper.

d. In the second experiment results, it would be informative to know whether participants in any condition were more likely to change their morality and labeling judgments after learning of the investment outcome. This could be accomplished by calculating a change score (reconsidered morality judgment – initial morality judgment), or by using another method. At the least, it would be useful to know whether the initial judgments varied at all by condition.

We have now introduced the change score as suggested by Reviewer 2. We are very grateful for this suggestion as it indeed leads to the novel finding that respondents actively downgrade investments if they turn out to result in a bad outcome more strongly than they upgrade investments if they turn out to have a good outcome. We elaborate on this finding in the revised version of our discussion and conclusion.

1. The study’s abstract only describes the first experiment, yet the study’s title and write-up contemplate both experiments. I recommend revising the abstract to reflect the inferences drawn from the second experiment. Likewise, the conclusion paragraph does not mention the inferences of experiment 2.

This objection is justified. We have therefore completely rewritten the abstract and added the consideration of study 2 requested by reviewer 2. Furthermore we adjusted the conclusion part.

2. Footnote 1 cites German Criminal Code (§46 paragraph 2 StGB), but does not explain what that part of the Code requires. More explanation would be helpful for the reader.

We have changed the example for better explanation.

3. The study is partially motivated through the idea that certain investment opportunities are often touted as being more ethical – investments with underlying focus on CSR, green initiatives, etc. However, there is a bit of disconnect between this motivation and what is operationalized in the study. The investment security in the study is described as a complex financial instrument with a value that fluctuates based on changes in oil prices. No mention is made of whether the underlying investment is geared toward an ethical cause. The authors could better describe this design choice in light of the overall goals of the research and how the research is motivated.

We have taken this important objection of Reviewer 2 seriously and added further comments to the mentioned paragraph.

4. The authors describe investment decisions, generally, as decisions made under conditions of Knightian uncertainty. I believe this is too strong of a statement. Typically, we think of investors making (or at least trying to make) rational decisions based on some estimation of expected value of the investment – i.e., managing risk as opposed to true Knightian uncertainty. While some investment contexts will reflect this high level of uncertainty, not all will. This also disconnects with the operationalization of the context in the experiment. In the vignettes, the investor is described as having consulted with financial advisors, presumably as a rational investor weighing risk and potential reward.

As mentioned above, we have slightly reworded the sentence and weakened the argument with uncertainty in the sense of Frank Knight. The objection of Reviewer 2 cannot be completely eliminated in our eyes. However, since this argument is not of greatest relevance for the following argumentation in the paper, we have rewritten the paragraph.

5. In the conclusion, the inferences of the study are extrapolated to infer that sustainability rating professionals may succumb to the moral luck bias when providing opinions regarding the moral evaluation of particular investments; however, this may not be the case. In the experiments, non-professionals make judgments demonstrating the moral luck effect. It is not clear that these results would generalize to professionals dedicated to such evaluations.

We have also considered this comment and included some comments.

 

Reviewer #3:

Thank you very much for the constructive comments on our manuscript. We think that our manuscript has very much benefitted from incorporating the suggestions you made in your referee report. In what follows, we will address each of your comments individually.

In study 1, they found people rate higher immoral scores for a "loss" scenario than a "profit" scenario, similarly there are less people who believe it is an investment in the "loss" scenario" than the "profit scenario". Study 2 was to analyze whether the observed pattern in Study 1 is based on moral reasoning or an unreflected moral intuition. 1. The literature has already showed people exhibit outcome bias (related to moral luck). I think the authors really need to emphasize how their work contribute to the literature. For instance, for study 1, even without their study, at least some readers would already think that people would base their evaluations and interpretations on outcome.

This objection of Reviewer 3 is justified. We have tried to supplement and enhance the argumentation with some comments at the mentioned point.

2. The section on “the concept of moral luck” is a bit too long and nearly included an entire paragraph of the original words from Kant. I don’t think it is necessary. It would make the story to stay on track better.

We thank Reviewer 3 for this objection and have removed the lengthy quote from Kant at this point.

3. Figure 2a appear to show the results after they see the outcome. I think readers would be interested to see how their ratings are immediately after they see only the decision but before the outcome is revealed.

We now report these numbers and conduct further testing. The information requested here by Reviewer 3 is now shown in the paper.

4. Figure information can be better presented. For instance, the authors can easily include a legend name of what each color of the bar represent. The authors could also consider showing the p-values on the graph (or the show the ** on the graph). Moreover, please provide a name for the figure, at the first glance Figure 1a and Figure 2a appear to be the same.

We have revised the graphs as requested and do now provide figure names.

---

## [Decision Letter · Decision Letter 1]

23 Aug 2022

PONE-D-22-02964R1Morally Unlucky Investors: We find lossy investments less moral, even upon reflectionPLOS ONE

Dear Dr. Max,

Thank you for submitting your manuscript to PLOS ONE. After careful consideration, we feel that it has merit but does not fully meet PLOS ONE’s publication criteria as it currently stands. Therefore, we invite you to submit a revised version of the manuscript that addresses the points raised during the review process.

Note that the reviewers have some minor comments. Consider them to improve the paper.

We look forward to receiving your revised manuscript.

Kind regards,

Alfonso Rosa Garcia

Academic Editor

PLOS ONE

Journal Requirements:

Reviewers' comments:

Reviewer's Responses to Questions

**Comments to the Author**

1. If the authors have adequately addressed your comments raised in a previous round of review and you feel that this manuscript is now acceptable for publication, you may indicate that here to bypass the “Comments to the Author” section, enter your conflict of interest statement in the “Confidential to Editor” section, and submit your "Accept" recommendation.

Reviewer #1: (No Response)

Reviewer #2: (No Response)

Reviewer #3: (No Response)

2. Is the manuscript technically sound, and do the data support the conclusions?

Reviewer #1: (No Response)

Reviewer #2: Partly

Reviewer #3: Yes

3. Has the statistical analysis been performed appropriately and rigorously? 

Reviewer #1: (No Response)

Reviewer #2: Yes

Reviewer #3: Yes

4. Have the authors made all data underlying the findings in their manuscript fully available?

Reviewer #1: (No Response)

Reviewer #2: Yes

Reviewer #3: (No Response)

5. Is the manuscript presented in an intelligible fashion and written in standard English?

Reviewer #1: (No Response)

Reviewer #2: Yes

Reviewer #3: Yes

6. Review Comments to the Author

Reviewer #1: I thank the authors for addressing some of my concerns. I would find it helpful if authors could highlight the changed passages in the manuscript or at least provide line numbers and reference them. The following concerns have in my view not been fully addressed:

⁃definition of “moral luck”: I guess the authors have been trying to rephrase Nelkin’s sentence “Moral luck occurs when an agent can be correctly treated as an object of moral judgment despite the fact that a significant aspect of what she is assessed for depends on factors beyond her control.” Nelkin states that the agent is *being judged*. In the manuscript, the authors give the impression that it is the agent who is *making a judgement*. This may not be intended but readers can easily get this impression. Please revise.

⁃There is still no rationale given for the chosen sample sizes. Please add these.

⁃Study 2: I do not understand why the “change score” removes the worry about initial differences in pre-evaluations between groups. Please add information about these.

⁃Please add tables for one-way ANOVAs for both studies for sake of completeness

⁃As far as I see, there is still no main effect of outcome in Study 2. Please acknowledge this.

⁃Williams reference: Thanks, now I see where the talk of moral luck as an oxymoron is coming from. Williams (“Moral Luck”, 1981) is a locus classicus for the term and Williams is generally regarded as being supportive of moral luck, so the way the term is introduced currently gives the misleading impression that Williams thought moral luck was an oxymoron. Instead, he wanted to draw attention to a puzzle. Please adjust this.

⁃Tables 1 and 2 contain three sub-columns (under “profit”) for “bank”, shouldn’t two of these be “company” and “municipality”?

Reviewer #2: Review of Manuscript # PONE-D-22-02964R1

Title: Morally Unlucky Investors: We find lossy investments less moral, even upon reflection

General Comment: The authors have addressed many of the comments I made in the previous round. This resulted in a different interpretation of the results of the second experiment, so the paper has changed somewhat. In what follows, I will re-summarize the paper as I see it and then provide further suggestions for refining the manuscript.

Summary:

This paper reports the results of two experiments conducted to examine two questions: (1) Do individuals evaluate the morality of investment decisions through a lens of moral luck, and (2) If so, is this bias conscious or unconscious?

In the first experiment, participants learn that an investment decision was made and that the investment was later determined to be either profitable or unprofitable (manipulated between subjects). Participants evaluate the decision to invest as being more immoral when the investment is unprofitable than when it is profitable. Participants were also more likely to label the profitable investment as an “investment” as opposed to “speculation” or “gambling.” These results are consistent with the concept of moral luck.

In the second experiment, participants learn about the investment decision and then assess the morality of the investment and the appropriate label before learning whether the investment was profitable or not. Participants then learn about the investment’s outcome (profitable or unprofitable, manipulated between subjects) and can revise their judgments on morality and labeling. Results indicate that participants revise their initial judgments more in the unprofitable condition. Specifically, the investment decision is reassessed as more immoral in the unprofitable condition than in the profitable condition. These results suggest that the bias toward finding profitable investments more moral is conscious.

Comments on Methods & Analyses:

1.) Tables:

a.For descriptive statistics (tables 1& 2), it would be helpful to see the means for gain, loss, and overall because the main analysis collapses across the context factor. That is – the authors are most interested in the comparison of the profit and loss conditions. The way the descriptives are currently presented seems to emphasize the context factor (bank, automotive company, and municipality) but doesn’t provide the means of most interest (profit and loss). The tables should also include the sample size, n, for each cell.

i.There is an error in labeling the conditions in these tables (bank, bank, bank…)

ii.I recommend the authors find a similar paper published in the journal and ensure that tables are formatted similarly.

b.The authors should include a table of descriptives for the change measures from experiment 2. A table for demographics should also be included (as it was for Study 1).

2.)The results and implications of study two are not well described. Please expound on what the analyses mean in terms of the overall research question addressed by Study 2: Is the moral luck bias in the investment context conscious or unconscious?

Other Comments:

1.) I do not believe the title of the paper appropriately describes the research.

a.The paper is not about “morally unlucky investors,” per se. Rather, the paper investigates moral luck in the investment context.

b.The word “lossy” is not typically used in the investment context. It has specific meaning associated with electrical conduction. I recommend the authors choose one way of describing the two possible investment outcomes (perhaps, “profitable” vs “unprofitable”) and use that terminology in the title and consistently throughout the paper.

2.)I recommend the authors carefully proofread and consider having the paper professionally copyedited prior to publication.

3.)Krische (2019) is cited to justify the participants in the study. However, Krische used a sample of all MTurk workers, while this study’s sample is drawn from a CloudResearch Prime Panel. Notably, the CloudResearch sample seems to be significantly older than the sample drawn by Krische. I recommend de-emphasizing Krische as support for the participants and better describing any filters that were used to screen participants recruited from CloudResearch.

Reviewer #3: Referee Report for PONE-D-22-02964R1

Comments:

1.Some simple explanation regarding why the treatment assignment is not symmetric is needed (when we read Table 1)

For instance, we have all categories of context (bank, auto and municipality) for loss, but only bank for profit. Since the authors says it is 2 x 3, I am wondering is it a typo? Sam for Table 2.

2.Table 3 (the authors on page 12 referred to Table 3a, which I believe is a typo)

There is a 10% significance for context, I am wondering whether the authors intend to discuss this part.

3.Table 4

We see a significant interaction between outcome and order, if so, wouldn’t it be cleaner to just analyze the data that comes first? I think the authors shall provide some explanations for this.

Other minor comment

I am slightly confused which part of the paper I shall read. As I was reviewing it, I noticed that there is another version of the paper again from Page 42. I think in the future, it might be easier not to include the original submission to avoid such confusion (but I understand it could be part of the journal’s requirement).

A kindly future suggestion to the authors:

This response to reviewer can be provided with more details to facilitate the review process. For instance, the authors can provide more details regarding the exact revisions they have made. And mention where these revisions are, on which page, which section?

7. PLOS authors have the option to publish the peer review history of their article (what does this mean?). If published, this will include your full peer review and any attached files.

Reviewer #1: No

Reviewer #2: No

Reviewer #3: No

---

## [Author Response · Author response to Decision Letter 1]

3 Oct 2022

Dear editor,

Thank you very much for the positive assessment of our revised manuscript. In the previous version, we have taken care of all the remaining comments by the reviewers. We also had our manuscript professionally proofread again, which resulted in numerous linguistic improvements.

We are looking forward to hearing from you.

Your sincerely,

The authors

 

Reviewer #1:

Thank you very much for the constructive comments on our manuscript. We think that our manuscript has very much benefitted from incorporating the suggestions you made in your referee report. In what follows, we will address each of your comments individually.

⁃ definition of “moral luck”: I guess the authors have been trying to rephrase Nelkin’s sentence “Moral luck occurs when an agent can be correctly treated as an object of moral judgment despite the fact that a significant aspect of what she is assessed for depends on factors beyond her control.” Nelkin states that the agent is *being judged*. In the manuscript, the authors give the impression that it is the agent who is *making a judgement*. This may not be intended but readers can easily get this impression. Please revise.

We rephrased the respective sentence to avoid the misunderstanding described by the reviewer.

⁃ There is still no rationale given for the chosen sample sizes. Please add these.

We added information on a power analysis in footnote 1 of the revised version of the paper.

⁃ Study 2: I do not understand why the “change score” removes the worry about initial differences in pre-evaluations between groups. Please add information about these.

We added footnotes 3 and 5 to provide the required information which shows that initial differences in pre-evaluations and pre-labels were not statistically significant.

⁃ Please add tables for one-way ANOVAs for both studies for sake of completeness

Tables for one-way ANOVAs have been added to Study 1 and Study 2.

⁃ As far as I see, there is still no main effect of outcome in Study 2. Please acknowledge this.

As per request of reviewer 2, we do now focus on the change score as our dependent variable in Study 2. As our new ANOVAs show, there is a main effect of outcome on the change score. There is, however, still no main effect of outcome on the label change of the investment. We now acknowledge this in the limitations of our study.

⁃ Williams reference: Thanks, now I see where the talk of moral luck as an oxymoron is coming from. Williams (“Moral Luck”, 1981) is a locus classicus for the term and Williams is generally regarded as being supportive of moral luck, so the way the term is introduced currently gives the misleading impression that Williams thought moral luck was an oxymoron. Instead, he wanted to draw attention to a puzzle. Please adjust this.

Thank you for pointing this out. We rephrased the respective passage to make this clear now.

⁃ Tables 1 and 2 contain three sub-columns (under “profit”) for “bank”, shouldn’t two of these be “company” and “municipality”?

Thank you for pointing this out. The table was changed according to a request by Reviewer 2.

 

Reviewer #2:

Thank you very much for the constructive comments on our manuscript. We think that our manuscript has very much benefitted from incorporating the suggestions you made in your referee report. In what follows, we will address each of your comments individually.

Tables: a. For descriptive statistics (tables 1& 2), it would be helpful to see the means for gain, loss, and overall because the main analysis collapses across the context factor. That is – the authors are most interested in the comparison of the profit and loss conditions. The way the descriptives are currently presented seems to emphasize the context factor (bank, automotive company, and municipality) but doesn’t provide the means of most interest (profit and loss). The tables should also include the sample size, n, for each cell. i. There is an error in labeling the conditions in these tables (bank, bank, bank…) ii. I recommend the authors find a similar paper published in the journal and ensure that tables are formatted similarly. b. The authors should include a table of descriptives for the change measures from experiment 2. A table for demographics should also be included (as it was for Study 1).

Thank you very much for this helpful remark. We have changed the tables accordingly and do now focus on the outcome conditions of profit and loss which are indeed at the core of our interest. Sample sizes were added and the formatting was checked. A table of descriptives for the change measures from Experiment 2 and a table for demographics have been included. 

The results and implications of study two are not well described. Please expound on what the analyses mean in terms of the overall research question addressed by Study 2: Is the moral luck bias in the investment context conscious or unconscious?

We added a clearer interpretation of this finding in the second paragraph of the conclusion of the paper where we discuss the results of Study 2.

I do not believe the title of the paper appropriately describes the research. a. The paper is not about “morally unlucky investors,” per se. Rather, the paper investigates moral luck in the investment context. b. The word “lossy” is not typically used in the investment context. It has specific meaning associated with electrical conduction. I recommend the authors choose one way of describing the two possible investment outcomes (perhaps, “profitable” vs “unprofitable”) and use that terminology in the title and consistently throughout the paper.

We appreciate this comment very much and have altered the title according to your suggestion. It now reads as follows: Moral Luck in Investment Contexts - We Consciously Find Unprofitable Investments Less Moral.

I recommend the authors carefully proofread and consider having the paper professionally copyedited prior to publication.

We sent the article to a proofreading agency and had the language revised. 

Krische (2019) is cited to justify the participants in the study. However, Krische used a sample of all MTurk workers, while this study’s sample is drawn from a CloudResearch Prime Panel. Notably, the CloudResearch sample seems to be significantly older than the sample drawn by Krische. I recommend de-emphasizing Krische as support for the participants and better describing any filters that were used to screen participants recruited from CloudResearch

This hint is very helpful. We have removed the reference to Krische from the article and added information on the filters that were used to screen participants.

Reviewer #3:

Thank you very much for the constructive comments on our manuscript. We think that our manuscript has very much benefitted from incorporating the suggestions you made in your referee report. In what follows, we will address each of your comments individually.

Some simple explanation regarding why the treatment assignment is not symmetric is needed (when we read Table 1) For instance, we have all categories of context (bank, auto and municipality) for loss, but only bank for profit. Since the authors says it is 2 x 3, I am wondering is it a typo? Sam for Table 2.

This was indeed a typo that is corrected in the present version of the manuscript. 

Table 3 (the authors on page 12 referred to Table 3a, which I believe is a typo)

Thank you for pointing this out. We have corrected this typo.

We see a significant interaction between outcome and order, if so, wouldn’t it be cleaner to just analyze the data that comes first? I think the authors shall provide some explanations for this.

Please notice that there is in fact no interaction effect between outcome and the order in which the questions where presented, but rather between outcome and whether the ex-ante label was investment or speculation and gambling.

Other minor comment: I am slightly confused which part of the paper I shall read. As I was reviewing it, I noticed that there is another version of the paper again from Page 42. I think in the future, it might be easier not to include the original submission to avoid such confusion (but I understand it could be part of the journal’s requirement).

We would like to sincerely apologize for the confusion this has caused and have ensured that this did not happen in the present submission.

A kindly future suggestion to the authors: This response to reviewer can be provided with more details to facilitate the review process. For instance, the authors can provide more details regarding the exact revisions they have made. And mention where these revisions are, on which page, which section?

All changes in the manuscript are now tracked.

---

## [Decision Letter · Decision Letter 2]

8 Nov 2022

PONE-D-22-02964R2Moral Luck in Investment Contexts: We Consciously Find Unprofitable Investments Less MoralPLOS ONE

Dear Dr. Max,

Thank you for submitting your manuscript to PLOS ONE. After careful consideration, we feel that it has merit but does not fully meet PLOS ONE’s publication criteria as it currently stands. Therefore, we invite you to submit a revised version of the manuscript that addresses the points raised during the review process.

There are some mistakes that need to be solved, as well as some points that need to be clarified, as pointed by Reviewer 3. Please, consider carefully her/his suggestions. 

We look forward to receiving your revised manuscript.

Kind regards,

Alfonso Rosa Garcia

Academic Editor

PLOS ONE

Journal Requirements:

Reviewers' comments:

Reviewer's Responses to Questions

**Comments to the Author**

1. If the authors have adequately addressed your comments raised in a previous round of review and you feel that this manuscript is now acceptable for publication, you may indicate that here to bypass the “Comments to the Author” section, enter your conflict of interest statement in the “Confidential to Editor” section, and submit your "Accept" recommendation.

Reviewer #2: All comments have been addressed

Reviewer #3: (No Response)

2. Is the manuscript technically sound, and do the data support the conclusions?

Reviewer #2: Yes

Reviewer #3: (No Response)

3. Has the statistical analysis been performed appropriately and rigorously? 

Reviewer #2: Yes

Reviewer #3: (No Response)

4. Have the authors made all data underlying the findings in their manuscript fully available?

Reviewer #2: Yes

Reviewer #3: (No Response)

5. Is the manuscript presented in an intelligible fashion and written in standard English?

Reviewer #2: Yes

Reviewer #3: No

6. Review Comments to the Author

Reviewer #2: The authors have addressed all of my comments. Nice work! .

Reviewer #3: The authors really need to read their paper before their submission, in this submission, Table 2 has a table that is completely empty. And the table below it is also broken by pages.

In one of my comment (maybe I said Table 4, but it is Table 5 this time) it did show "outcome x order" has a significant effect. I don't think the authors have addressed my comments and barely responded to it. I read what is on their table. If it was a misunderstanding, then the authors need to clarify it in in depth.

Moreover, the tables generally can be formatted a bit better (please refer to some other publications either on plosone or other journals).

- e.g. spaces between figures/tables and text do not appear to follow any systematic pattern (e.g. see those above Figure 1a's title)

- tables just generally appear to be a bit like raw presentation. Some of the boarders looks thicker while others are thinner.

7. PLOS authors have the option to publish the peer review history of their article (what does this mean?). If published, this will include your full peer review and any attached files.

Reviewer #2: No

Reviewer #3: No

---

## [Author Response · Author response to Decision Letter 2]

18 Nov 2022

Reviewer #3:

We are grateful for Reviewer 3’s careful reading of our manuscript and apologize that we did not address all concerns in the previous revision of our manuscript. We are confident that the current revision takes care of the reviewer’s remaining concerns. In what follows, we address each remaining concern individually.

Reviewer #3: The authors really need to read their paper before their submission, in this submission, Table 2 has a table that is completely empty. And the table below it is also broken by pages.

ANSWER: We are very sorry for this mistake. We have deleted the empty table and corrected the broken table.

In one of my comment (maybe I said Table 4, but it is Table 5 this time) it did show "outcome x order" has a significant effect. I don't think the authors have addressed my comments and barely responded to it. I read what is on their table. If it was a misunderstanding, then the authors need to clarify it in in depth.

ANSWER: We apologize for not having addressed this comment sufficiently. In the revised version of the manuscript, we follow the previous suggestion of Reviewer 3 and perform an additional analysis that focuses on the condition where the labeling of the decision was done first. This analysis is now summarized in Table 6 which is discussed on p. 15 of the revised version of the manuscript. It indeed turns out that the influence of the investment’s outcome no longer predicts the labeling of the decision, if only those respondents are considered who label the investment before it is evaluated.

Moreover, the tables generally can be formatted a bit better (please refer to some other publications either on PLOS ONE or other journals).

- e.g. spaces between figures/tables and text do not appear to follow any systematic pattern (e.g. see those above Figure 1a's title)

- tables just generally appear to be a bit like raw presentation. Some of the boarders looks thicker while others are thinner.

ANSWER: We referred to other PLOS ONE publications where the tables have an upper and a lower frame as well as inner frames. Tables are now formatted consistently and spaces between figures and tables have been checked for consistency as well.

---

## [Editor Report · Decision Letter 3]

22 Nov 2022

Moral Luck in Investment Contexts: We Consciously Find Unprofitable Investments Less Moral

PONE-D-22-02964R3

Dear Dr. Max,

We’re pleased to inform you that your manuscript has been judged scientifically suitable for publication and will be formally accepted for publication once it meets all outstanding technical requirements.

Kind regards,

Alfonso Rosa Garcia

Academic Editor

PLOS ONE
---

## [Editor Report · Acceptance letter]

1 Dec 2022

PONE-D-22-02964R3 

Moral Luck in Investment Contexts:
We Consciously Find Unprofitable Investments Less Moral 

Dear Dr. Max:

I'm pleased to inform you that your manuscript has been deemed suitable for publication in PLOS ONE. Congratulations! Your manuscript is now with our production department. 

Kind regards, 

on behalf of

Dr. Alfonso Rosa Garcia 

Academic Editor

PLOS ONE